# School and home-based educational intervention in urban Kenya: Sustained improvements in knowledge, attitudes, and practices for *Aedes aegypti* control

Prathik Kalva[1,2]*, Jenna Forsyth[3], Francis Mutuku[4], Gladys Agola[4], Mwashee Lutt[4], Angelle Desiree LaBeaud[1]

1 Department of Pediatrics, Stanford University School of Medicine, Stanford, California, United States of America, 2 School of Medicine, Baylor College of Medicine, Houston, Texas, United States of America, 3 Stanford Woods Institute for the Environment, Stanford University, Stanford, California, United States of America, 4 Technical University of Mombasa, Mombasa, Kenya

* prathik.kalva@gmail.com

## Abstract

*Aedes aegypti* mosquitoes are the primary vectors for important arboviral health threats including chikungunya (CHIKV) and dengue (DENV) viruses, primarily breeding in domestic plastic containers. Previous studies have highlighted a severe lack of knowledge about non-malarial mosquito-borne diseases in Kenya, so we proposed a household and school-based educational intervention in urban coastal Kenya to determine whether it could bring about long-term improvements in knowledge, attitudes, and practices related to the source reduction of arboviral disease vectors. In this cluster-randomized controlled trial, 249 households from 5 villages were placed in the intervention arm and 243 households from 5 other similar villages were placed in the control arm. From each household, one fourth grade child was enrolled. Data on the child participants' knowledge, attitudes, and practices (KAP) were collected at baseline and 3- and 12-months post-intervention, along with counts of immature mosquitoes in containers in the participants' households. At 3 months, the intervention group showed significantly greater improvement in attitude scores compared to the control group (p = 0.01), with no significant differences in knowledge or practices. However, by 12 months, the intervention group demonstrated significant improvements in knowledge (1.51 vs. 0.35, p < 0.001), attitude (0.268 vs. −0.263, p < 0.001), and practice (0.118 vs. −0.235, p < 0.001) scores compared to the control group. Additionally, the house index (houses with at least one productive container) increased in both the intervention (13.9% to 25.6%) and control (9.4% to 28.4%) households, signifying that although the intervention improved knowledge and some practices, a more targeted approach is needed to address challenges in vector control. This study demonstrates that long-term advances in knowledge, attitudes,

**Data availability statement:** All relevant data are within the article and its supporting information files.

**Funding:** This study was financially supported by Stanford University's Child Health Research Institute Bechtel Faculty Scholar Award granted to A.D.L. and Stanford's Emmett Interdisciplinary Program in Environment and Resources, and the Center for African Studies Awards granted to J.F. The funders had no role in the study design, data collection and analysis, decision to publish, or preparation of the manuscript.

**Competing interests:** The authors have declared that no competing interests exist.

and practices regarding arboviral diseases can be achieved through household and school-based educational interventions.

## Introduction

Arthropod-borne viruses have emerged as global pathogens and have presented themselves as a substantial threat to both human and animal communities [1–4]. In the past decade, there has been an increase in the number of arboviral disease cases, in part due to chikungunya (CHIKV) and dengue (DENV) virus outbreaks [5]. Specifically in Kenya, there have been large-scale outbreaks of CHIKV [6] and DENV [7], which have caused a great deal of social and economic harm to both urban and rural communities [8]. The *Aedes aegypti* mosquito is the main vector for CHIKV and DENV and is a very efficient transmitter of these arboviral diseases [8] and likely to expand due to climate change [9,10].

Globally, dengue is one of the most prevalent and rapidly spreading arboviral diseases, with an estimated 390 million infections annually. While historically considered a tropical and subtropical illness, recent evidence suggests that Africa, including sub-Saharan Africa, is a new hotspot and bears a significantly underestimated dengue burden due to limited surveillance and frequent misdiagnosis as malaria or another febrile illness [11]. Dengue can cause a range of clinical presentations including mild febrile illness to severe dengue shock syndrome, which has a mortality rate of less than 1% with proper treatment but can be greater than 20% in the absence of treatment [12]. Children under five and infants, in particular, are at a higher risk of developing serious dengue likely due to poor immune response and antibody-dependent enhancement [11,13]. Chikungunya virus is another arboviral disease associated with substantial morbidity. Although its mortality rate is lower than that of severe dengue, CHIKV infection typically cause acute febrile illness which can be accompanied by severe myalgias, leading to chronic arthralgia in up to 40% of cases, severely limiting quality of life [11,14].

There has been a growing public health threat posed by DENV and CHIKV in Kenya due to several factors including rapid urbanization, socioeconomic disparities, plastic pollution, and poor sanitation [11,14,15]. Furthermore, Kenyan healthcare facilities often lack the point-of-care diagnostic tools needed for DENV and CHIKV screening, further contributing to poor healthcare outcomes [11]. Finally, the lack of a standardized reporting system and overlapping clinical presentations between DENV and other endemic febrile illnesses (typhoid and malaria) all contribute to the increased disease burden in Kenya [11]. Expanding surveillance and introducing targeted intervention strategies are crucial for mitigating the impact of these often overlooked diseases.

Although numerous anti-malaria campaigns have been conducted in Kenya [16], they have not been effective in stopping the spread of CHIKV and DENV because bednets are not effective against day-biting *Aedes aegypti* mosquitoes [17]. Additionally, adult *Aedes aegypti* mosquitoes are difficult to control because they are highly

anthropophilic [18], requiring only one tablespoon of water for successful ovipositioning, and they are highly resistant to insecticides, partly because they preferentially hide in human-made containers [19–23]. Multiple studies have highlighted the increased arboviral risk in urban communities due to increased mosquito presence, increased contact with humans, and decreased biodiversity [19,24–28]. Also, recent modeling predicts that climate change will lead to more *Aedes aegypti* mosquito habitats and prolong transmission seasons, potentially intensifying the risk in urban environments [11].

These factors combined make *Aedes aegypti* mosquitoes a severe threat to urban communities in Kenya.

Previous work performed in coastal Kenyan communities has shown a significant lack of knowledge regarding source reduction techniques for non-malaria arboviral diseases [29]. Larval source reduction involves the removal or treatment of mosquito breeding sites such as domestic house containers to disrupt the mosquito life cycle and prevent breeding. However, simply educating community members with printed materials and lesson plans on the importance of these arboviral diseases is not enough [30]. According to the World Health Organization, there needs to be significant involvement of trained experts who can educate and demonstrate simple source reduction techniques to engage communities in participation for more success in turning education into action [31]. In endemic regions, community led environmental clean-ups and integration of mosquito surveillance programs have worked to reduce *Aedes aegypti* populations and could serve as complementary interventions to educational programs [11].

Community-based educational interventions have been shown to be productive in improving knowledge and bringing out positive changes in attitudes and behaviors regarding arboviral diseases. For example, in India, health education-based interventions led to significant increases in knowledge regarding mosquito-borne diseases and an increase in self-reported personal protection measures [32].A review article on educational interventions specifically targeting *Aedes aegypti* found that they were successful in reducing entomological indicators [33]. However, despite these increases in knowledge, a study in Kenya showed that educational interventions alone may not always translate to strong and effective behavior changes and that more community engagement and outreach is needed to influence behavior changes [34].

School-based educational interventions have been effective in bringing about changes in knowledge and behaviors in children [35] because children are often seen as "agents of change" with a stronger willingness to adopt new source reduction techniques and express their views about a more positive future compared to adult caregivers [36]. In Cambodia [37] and Puerto Rico [38], school-based programs led to significant improvements in children's knowledge as well as the adoption of source reduction techniques related to DENV. However, the effectiveness of these school-based programs depends on the target population. A study in Malaysia showed that children in a flooding area had significant improvements in knowledge as well as behaviors but the children from a non-flooded area only had improvements in knowledge, demonstrating how risk factors and urgency plays a role in the translation of knowledge into behavior [35]. In a study conducted in rural Kenyan communities, it was seen that a combined household and school-based educational program brought about significant improvements in arboviral disease knowledge and source reduction behaviors in children [39], but whether a similar intervention could work in at risk urban environments was unknown.

This study aims to address this gap in understanding the effectiveness of a combined household and school-based educational intervention for *Aedes aegypti* control in urban communities. To the best of our knowledge, this is the first study to evaluate the impact of such an intervention in an urban setting. By targeting both children and caregivers, this approach seeks to create a more comprehensive and sustainable approach to vector control compared to single-setting interventions.

Our primary research question is whether a combined household and school-based educational intervention can significantly improve long-term knowledge, attitudes, and behaviors related to Aedes aegypti control in urban Kenya, and we hypothesize that it can. Our secondary research question is whether this intervention can lead to significant reductions in the number of immature Aedes aegypti larvae as well as potential breeding sites in households over time, and we hypothesize that it will be able to.

## Methods

### Ethical considerations

Adult caregivers provided written consent for participating students and themselves. In addition, we obtained assent from every child participant aged 7 and older. The study protocol was reviewed and approved by the ethical review committee at the Kenyatta National Hospital/University of Nairobi (protocol # 241/03/2016) and the Institutional Review Board (IRB) of Stanford University (protocol #35504).

Additional information regarding the ethical, cultural, and scientific considerations specific to inclusivity in global research is included in the S1 Text.

### Study area

This study was conducted in the Likoni sub-county of Mombasa County in Kenya. Likoni was selected because of its high populational density (250,358 people living in an area of 40.5 sq. km), urban characteristics, and previous reports of dengue outbreaks and a 2013 study showing a 13% prevalence of current or recent DENV infection [40]. The main villages in Likoni are Mrima, Longo, Maji Safi, and Mweza. This area experiences two rainy seasons: one long rainy season from March-June and a short rainy season from October to December. In the Likoni division there is an average annual rainfall of 997 mm with a mean temperature of 26 °C.

### Intervention design and piloting

The intervention design and piloting for this study follows the same guidelines as described previously [39]. The pilot study was conducted to refine the intervention materials and methods, ensuring they were culturally appropriate and effective. The pilot involved a smaller sample of households and schools (150 children and 150 caregivers across three villages) to test the feasibility of the intervention. Feedback from participants was used to make adjustments before full-scale implementation.

### Village selection

Villages were selected based on their urban characteristics, including population density, infrastructure (distance from a main road), and accessibility. Initially, 25 villages were examined for eligibility, and 15 were excluded based on the following exclusion criteria: i) located more than 5km from a main road or impassable during rainy season (n = 7), ii) the village public primary school had participated in a research project or the pilot study for this intervention (n = 6), and iii) the village had no suitable pair village of similar urban status within 5 km to serve as control (n = 2). The remaining 10 villages were then paired based on proximity and the urban status of their communities: 5 villages were randomly assigned to the intervention arm, and 5 were assigned to the control arm.

### Participants

The study included fourth grade students and their primary caregivers. Fourth-grade students were chosen because they are at an age where they can understand and engage with the educational content and they are often seen as "agents of change" within their households. Each village's public primary school participated, and within each village, 60 children were randomly selected, along with their primary caregivers.

### Intervention implementation

A cluster-randomized controlled trial was used, with each village considered a cluster. The intervention villages were in the Likoni division of Kenya, and they received this intervention between May 2019 and June 2019. The impact of the educational intervention was observed through the rainy season (March-June), which corresponds to the peak transmission

season in the monsoonal climate. The baseline survey was given in May/June 2019, the 3-month follow-up was given in August/September 2019, and the 12-month follow-up was given in May/June 2020. At the end of the year-long study once the final data was collected from the intervention villages, the students in the control villages also received the educational intervention.

The intervention was implemented by two trained research team members who delivered the curriculum daily for one hour in an interactive after-school session for five consecutive days in participating schools (one week). A similar intervention with two trained research team members was also given to the children's primary caregivers at their homes through 1-hour educational sessions. A control intervention targeting hand washing was administered to control school classrooms and caregivers. The trainers spoke the same language as the caregivers and children to ensure effective communication.

## Curriculum development and components

A curriculum committee met to discuss the intervention curriculum and included representatives from the Likoni Ministry of Health (MOH), MOH vector-borne disease control unit, school teachers, and various local hospitals. After taking varied perspectives into consideration and using recommendations from the health belief model, the final curriculum for both the school-based interventions was created. The main goal of the educational portion of the curriculum was to make the children and caregivers aware of the fact that mosquitoes other than simply the *Anopheles* mosquitoes (malaria) are of concern. Furthermore, we wanted to emphasize how dangerous standing water can be and relay the importance of turning over unused water containers to get rid of potential mosquito breeding sites. The behavioral portion of the curriculum focused on teaching source reduction techniques such as covering water containers or dumping out standing water. In this portion of the curriculum, we used active learning that included demonstration and practice of source reduction techniques to promote learning and behavior change [41]. In addition, we created and included a novel mosquito-tag game for the children to play with the goal of teaching the children three main concepts: i) the mosquito lifecycle and the importance of rainwater, ii) how containers that hold water and are not sealed can be places where mosquitos lay eggs, and iii) the role of humans in controlling mosquito breeding. We also had a household mapping session where trained individuals visited households to help caregivers identify containers within their house that could be susceptible to becoming breeding sites for mosquito larvae. The goal of these interactive intervention strategies was to promote hands-on learning and allow the caregivers and children to apply the knowledge that they learned.

As part of the intervention, we also provided handouts to the caregivers and children so that they could easily access the information that they were taught as well as to encourage discourse about what they had learned (S1 Fig). For children and caregivers, particular focus was given to picking up and re-using containers that had no purpose, specifically plastic containers because the children could then use them as part of a school-based seed planting competition. For caregivers, we also added information about covering or emptying containers with a purpose such as those used for laundry or sanitation.

## Vector assessment

At baseline and 12 months, an entomological survey was administered to the households to measure vector data related to potential mosquito breeding sites at the homes and risk factors. This survey was administered during the rainy season at both baseline and the 12-month follow-up. Firstly, the survey identified and mapped water containers around the house, and classified the containers based upon their purpose, such as drinking water, laundry, cooking, bathing, animal water, or other. Additionally, the survey assessed household members' perceptions of each container as either harmful (potential mosquito breeding site) or beneficial (essential for daily use). Furthermore, the size and source of water for each container was documented as well as whether each container was shaded or covered. Finally, careful inspection of each container for immature mosquitos (larvae or pupae) was conducted and the number of immature mosquitos within each container was noted to assess the productivity of different container types for breeding.

## Intervention evaluation

At baseline, demographic characteristics were assessed for the participants in the control and intervention arms. Furthermore, at baseline, 3 months, and 12 months, participants' knowledge, attitudes, and practices (self-reported and perceived) were assessed using a 34-question survey that focused on content covered during the intervention and open-ended questions.

For the open-ended questions, the knowledge question specifically asked was "What are the best ways to prevent mosquito breeding?" and the behavior question asked was "What do you do to protect yourself from mosquitoes?" Enumerators were trained to pose these questions verbatim, pause for them to answer, and then prompt them further by asking "anything else?" up to two additional times. Responses to these questions were recorded using a predefined list that included source reduction techniques and general mosquito control measures. Source reduction behaviors included those focused on in the intervention such as covering containers, disposing of trash and unused containers, moving containers to covered areas, or turning over containers. Participants who mentioned at least one of these behaviors were coded as "1" for analysis and the specific source reduction techniques they mentioned were recorded as well for further analysis. Those that did not mention any techniques or knowledge of arboviral disease were coded as "0." Attitudes were elicited from participants by asking them whether they felt a specific practice was important (coded as "1") or not (coded as "0") for preventing mosquito breeding.

Self-reported practices involved asking participants if they were engaging in a source reduction activity while a perceived practice asked if a participant felt that they had the ability to engage in a source reduction practice. Differences in primary outcomes such as knowledge, attitudes, and behaviors related to mosquito-borne diseases were compared between the intervention and control arms at baseline, 3- and 12-months post-intervention. Secondary outcomes such as vector data at households and risk factors were assessed and compared at baseline and after 12 months to maintain seasonality.

## Statistical analysis

All statistical analyses were conducted using R version 4.2.0 [42]. For the educational intervention data, a two-sample test for equality of proportions with continuity correction was employed to compare the proportion of correct responses to survey questions between the intervention and control groups. A significance level of 0.05 was applied to determine whether differences in the proportion of participants answering "correctly" between the two groups were statistically significant. Correct responses to the questionnaire were pre-determined based on the pilot study and aligned with the educational content of the intervention.

To further analyze the impact of the intervention, composite scores for knowledge, attitudes, and practices were developed for each participant. The knowledge score, with a maximum value of 4, was calculated based on whether participants correctly identified the need to cover water containers, remove trash and unused containers, understood that mosquitoes can transmit diseases, and recognized the specific mosquito responsible for malaria. The attitude score (ranging from 0 to 3) was determined by assigning a value of "1" for each of the following beliefs: that removing water containers is important, that water containers could be harmful, and that unused containers should be addressed. The practice score, ranging from 0 to 3, combined self-reported and perceived practices, with a score of "1" assigned for each of the following: perceiving the ability to remove unused containers, perceiving the ability to turn over unused containers, and reporting the practice of turning over sanitation containers. To evaluate the mean changes in these composite scores over time, each participant's change in knowledge, attitude, and practice scores was calculated by determining the difference between their baseline score and their score at 3 months or 12 months post-intervention. The mean of these individual changes was then computed for both the intervention and control groups, and a two-sample t-test was conducted to assess whether the intervention group demonstrated a significantly greater improvement compared to the control group. This approach helped control for any baseline differences between the groups.

For the vector household data, chi-square tests were performed to compare the proportions of immature mosquitoes in different container types and purposes in the control and intervention households. Correlation analysis was used to identify trends between different container types and purposes and immature mosquito breeding. Both tests were conducted using a significance level of 0.05.

## Exclusion criteria

Children who did not have data from all three time points (baseline, 3 months, and 12 months) were excluded from analysis of the KAP data. Similarly, households without vector data at both baseline and 12-month follow-up were excluded from analysis. Caregiver responses to the questionnaire were not analyzed due to differences at baseline (S1 Table).

Although excluding some participants could impact the statistical power of the study, the sample size was determined to ensure sufficient power to detect meaningful differences in primary outcomes. Any deviations from the intended sample size were carefully considered.

## Results

### Demographic information

At baseline, demographic information was collected from 600 caregivers but 108 were not included due to incomplete information, leaving a total of 492 caregivers: 243 in the control arm and 249 in the intervention arm (see Flow chart in S2 Fig). Similarly, information about their child that was in primary school (fourth grade) was also collected.

In the control arm, the mean age of the caregivers was 40 years (median: 39, range: 18–75) while in the intervention arm the mean age was 39 years (median: 38, range: 20–75) (Table 1). For the children, in both the control and intervention arms, the mean age was 12 years (range 9–16 years). The average number of people in the household was similar for both the control and intervention arms, which is important because with more people in a household, there is a larger number of water containers used by the family. The majority of the caregivers were Muslim (72% in the control arm and 75% in the intervention arm), and this is significant because it likely meant that in these households there were separate or additional water containers used for sanitation purposes. Although pit toilets were more common in control households (79%) compared to intervention households (68%), the control households had more assets (radio and television), signaling that the SES was comparable between the groups. Finally, 90% of participants in the control arm and 94% in the intervention arm used bednets to prevent mosquito-borne illnesses, signaling a similar knowledge base for mosquito-borne health threats (Table 1).

### Knowledge, attitudes, and practices (KAP) surveys

Overall, a total of 600 children completed the KAP surveys at baseline, but 445 children received and completed the KAP surveys at the 3- and 12-month follow-ups with 217 in the control arm and 228 in the intervention arm. (see Flow chart in S3 Fig).

At baseline, the control and intervention groups performed similarly in the knowledge, attitudes, and practices questions.

At the 3-month follow-up, the control and intervention groups performed similarly in the knowledge, attitudes, and practices (self-reported and perceived) questions (p > 0.05, Table 2). However, at the 12-month follow-up, a significant difference was observed in the proportion of correct answers by the intervention group compared to the control group in all three categories (p < 0.05, Table 2).

When examining the change in composite knowledge, attitude, and practice scores from baseline, the intervention group showed a significantly greater improvement in attitude compared to the control group at 3 months post-intervention (0.232 vs. -0.023, p = 0.01), but no significant differences were observed for changes in knowledge (0.535 vs. 0.295,

**Table 1. Baseline characteristics of participants in the control and intervention arms.**

| Baseline characteristics | Control | Intervention | P-Value |
|---|---|---|---|
| | (N = 243)[c] | (N = 249)[c] | |
| *Individual* | | | |
| Caregiver age (years)[a] | 40 (9.25) | 39 (9.05) | 0.23 |
| Caregiver sex (female) | 197 (81%) | 194 (78%) | 0.45 |
| Caregiver marital status (married) | 178 (73%) | 194 (78%) | 0.27 |
| Child age (years)[a] | 12 (1.08) | 12 (1.55) | 0.99 |
| Child sex (female) | 128 (53%) | 109 (44%) | 0.06 |
| *Household* | | | |
| Number of people per household | 5.9 (2.0) | 6.0 (2.1) | 0.59 |
| Religion (Islam) | 173 (72%) | 187 (75%) | 0.38 |
| Main source of water (well/borehole) | 3 (1%) | 8 (3%) | 0.24 |
| Boil water before drinking | 0 (0%) | 3 (1%) | NA |
| Toilet (pit toilet) | 192 (79%) | 170 (68%) | 0.01 |
| Has electricity | 220 (91%) | 222 (89%) | 0.72 |
| Owns a radio | 204 (84%) | 133 (53%) | <0.01 |
| Owns a television | 189 (78%) | 170 (68%) | 0.02 |
| *Mosquito-related* | | | |
| Use bednets | 219 (90%) | 234 (94%) | 0.16 |

[a]Mean (standard deviation)

[c]Percent calculated based on the number of respondents to the given survey (see S1 Fig).

p = 0.12) or practice (0.013 vs. -0.106, p = 0.27). However, by 12 months post-intervention, the intervention group significantly outperformed the control group in improvements in all three categories: knowledge (1.51 vs. 0.35, p < 0.001), attitude (0.268 vs. -0.263, p < 0.001), and practice (0.118 vs. -0.235, p < 0.001) (Table 3).

Although the KAP survey and open-ended questions were also asked to caregivers, their data was not analyzed due to strong differences in knowledge, attitudes, and practices at baseline between control and intervention caregivers (S1 Table).

### Vector data

**Baseline findings.** At baseline, the house index (percentage of households with productive containers) for the control households was 9.43% and 13.87% for the intervention households (Table 2).

The distribution of immature mosquitoes in the control group showed a high prevalence in drums (64.71%), followed by tires (17.90%), and buckets (11.13%). Containers with no specific purpose and those used for laundry were the primary breeding sites, contributing to 28.13% and 66.24% of immature mosquitoes, respectively (Fig 1).

In the intervention group, the distribution was more varied. Containers used for animal water had the highest percentage of immature mosquitoes (43.44%), followed by small containers (26.19%), and drums (15.16%). Containers with no purpose (6.64%) and those used for laundry (14.10%) also showed breeding activity (Fig 1).

When comparing the different breeding habitats, there was no significant difference in the distribution of immature mosquitoes among different container types ($\chi 2 = 5.23$, p = 0.26) or container purposes ($\chi 2 = 4.67$, p = 0.32) between the control and intervention groups at baseline. Through a correlation analysis, there was a moderate association between the presence of drums and the number of immature mosquitoes (r = 0.45, p < 0.01) in both the control and intervention groups at baseline (Fig 1).

**Table 2. Baseline, 3-month follow-up and 12-month follow-up knowledge, attitudes, and behavior for children in the control (n = 217) and intervention (n = 228) groups and entomological indices.**

| | BASELINE | | | 3 MONTHS | | | 12 MONTHS | | |
|---|---|---|---|---|---|---|---|---|---|
| | Control n (%) | Intervention n (%) | P-Value | Control n (%) | Intervention n (%) | P-Value | Control n (%) | Intervention n (%) | P-Value |
| **Knowledge** | | | | | | | | | |
| Know at least 1 source reduction technique | 116 (53%) | 126 (55%) | 0.77 | 151 (70%) | 176 (77%) | 0.09 | 144 (66%) | 209 (92%) | <0.001 |
| Cover water containers | 69 (32%) | 80 (35%) | 0.53 | 84 (39%) | 94 (41%) | 0.65 | 53 (24%) | 123 (54%) | <0.001 |
| Remove trash and unused containers | 50 (23%) | 49 (21%) | 0.78 | 77 (35%) | 95 (42%) | 0.21 | 103 (47%) | 131 (57%) | 0.04 |
| Know that can only get sick by mosquito bite | 124 (57%) | 131 (57%) | 1 | 130 (60%) | 137 (60%) | 1 | 157 (72%) | 198 (87%) | <0.001 |
| Know which type of mosquito causes malaria | 67 (31%) | 81 (36%) | 0.35 | 53 (24%) | 87 (38%) | 0.003 | 67 (31%) | 152 (67%) | <0.001 |
| **Attitude** | | | | | | | | | |
| Believe that removing water containers is important | 118 (54%) | 118 (52%) | 0.65 | 114 (53%) | 136 (60%) | 0.16 | 116 (53%) | 163 (71%) | <0.001 |
| Believe that water containers are harmful | 77 (35%) | 69 (30%) | 0.28 | 80 (37%) | 86 (38%) | 0.93 | 50 (23%) | 91 (40%) | <0.001 |
| Believe that unused containers need to be addressed | 142 (65%) | 137 (60%) | 0.29 | 139 (64%) | 155 (68%) | 0.44 | 114 (53%) | 134 (59%) | 0.22 |
| **Practices** | | | | | | | | | |
| *Perceived* | | | | | | | | | |
| Be able to remove unused water containers | 177 (82%) | 185 (81%) | 1 | 167 (77%) | 184 (81%) | 0.39 | 196 (90%) | 224 (98%) | <0.001 |
| Be able to turn over unused water containers | 170 (78%) | 173 (76%) | 0.61 | 161 (74%) | 178 (78%) | 0.39 | 195 (90%) | 225 (99%) | <0.001 |
| *Self-Reported* | | | | | | | | | |
| Turn over sanitation containers | 118 (54%) | 142 (62%) | 0.11 | 116 (53%) | 141 (62%) | 0.09 | 23 (11%) | 80 (35%) | <0.001 |
| **Entomological Indices** | | | | | | | | | |
| House Index (Houses with at least one productive container) | 15 (9%) | 19 (14%) | 0.70 | | | | 60 (28%) | 51 (26%) | 0.239 |

**Table 3. Mean change in composite knowledge, attitude, and practice scores from baseline to 3 months and 12 months post-intervention.**

| | 3 Months | | | 12 Months | | |
|---|---|---|---|---|---|---|
| | Control | Intervention | P-Value | Control | Intervention | P-Value |
| Knowledge | 0.295 | 0.535 | 0.12 | 0.35 | 1.51 | <0.001 |
| Attitude | −0.023 | 0.232 | 0.01 | −0.263 | 0.268 | <0.001 |
| Practice | −0.106 | 0.013 | 0.27 | −0.235 | 0.118 | <0.001 |

**12-month follow-up findings.** After 12 months, the house index for the control households was 28.44% and 25.63% for the intervention households (Table 2).

There were also notable changes observed in the distribution of immature mosquitoes. In the control group, drums continued to hold a significant proportion of immature mosquitoes (27.45%), although this was a decrease from baseline. Jerrycans saw a substantial increase, containing 38.93%, while buckets and small containers collectively accounted for 33.07% (Fig 1).

| Control Baseline | Bucket | Tire | Jerrycan | Small Containers | Drum | Other (AFC, Coconut, | Total |
|---|---|---|---|---|---|---|---|
| No purpose | 5.88% | 17.90% | 0.00% | 2.43% | 0.00% | 1.92% | 28.13% |
| Sanitation | 0.00% | 0.00% | 0.00% | 1.92% | 0.00% | 0.00% | 1.92% |
| Laundry | 3.20% | 0.00% | 0.00% | 0.00% | 63.04% | 0.00% | 66.24% |
| Animal water | 0.00% | 0.00% | 0.00% | 0.00% | 0.00% | 0.00% | 0.00% |
| Other | 2.05% | 0.00% | 0.00% | 0.00% | 1.66% | 0.00% | 3.71% |
| Total | 11.13% | 17.90% | 0.00% | 4.35% | 64.71% | 1.92% | 100.00% |

| Intervention Baseline | Bucket | Tire | Jerrycan | Small Containers | Drum | Other (AFC, Coconut, | Total |
|---|---|---|---|---|---|---|---|
| No purpose | 4.43% | 1.45% | 0.77% | 0.00% | 0.00% | 0.00% | 6.64% |
| Sanitation | 0.00% | 0.00% | 0.00% | 0.00% | 0.00% | 0.00% | 0.00% |
| Laundry | 4.51% | 0.00% | 0.00% | 0.00% | 9.58% | 0.00% | 14.10% |
| Animal water | 0.00% | 0.00% | 0.00% | 0.00% | 0.00% | 43.44% | 43.44% |
| Other | 2.85% | 0.00% | 1.19% | 26.19% | 5.58% | 0.00% | 35.82% |
| Total | 11.80% | 1.45% | 1.96% | 26.19% | 15.16% | 43.44% | 100.00% |

| Control 12 Months | Bucket | Tire | Jerrycan | Small Containers | Drum | Other (AFC, Coconut, | Total |
|---|---|---|---|---|---|---|---|
| No purpose | 0.00% | 0.57% | 0.00% | 0.56% | 0.00% | 0.00% | 1.12% |
| Sanitation | 0.00% | 0.00% | 0.00% | 0.00% | 0.00% | 0.00% | 0.00% |
| Laundry | 19.04% | 0.00% | 0.99% | 2.18% | 0.00% | 0.00% | 22.22% |
| Animal water | 0.00% | 0.00% | 0.00% | 0.00% | 0.00% | 0.00% | 0.00% |
| Other | 10.94% | 0.00% | 37.93% | 0.34% | 27.45% | 0.00% | 76.66% |
| Total | 29.99% | 0.57% | 38.93% | 3.08% | 27.45% | 0.00% | 100.00% |

| Intervention 12 Months | Bucket | Tire | Jerrycan | Small Containers | Drum | Other (AFC, Coconut, | Total |
|---|---|---|---|---|---|---|---|
| No purpose | 4.53% | 4.26% | 0.00% | 0.75% | 0.00% | 0.00% | 9.55% |
| Sanitation | 0.00% | 0.00% | 0.00% | 0.00% | 0.00% | 0.00% | 0.00% |
| Laundry | 1.56% | 0.00% | 5.62% | 0.00% | 0.00% | 0.00% | 7.18% |
| Animal water | 0.00% | 0.00% | 0.00% | 0.17% | 11.00% | 0.41% | 11.58% |
| Other | 15.02% | 0.00% | 34.62% | 1.01% | 21.04% | 0.00% | 71.69% |
| Total | 21.12% | 4.26% | 40.24% | 1.93% | 32.04% | 0.41% | 100.00% |

**Fig 1. Comparison of mosquito habitats by type and purpose at baseline and after 12 months among the control and intervention arms.** Percent of immature mosquitoes (larvae (early and late instars) and pupae) are reported within the cells of the table highlighted with green, yellow, orange, and red representing <1%, 1–4.99%, 5–19.99%, and >20% or larval abundance, respectively. The habitat types were distinguished according to size: 1) small containers (<5L), 2) tires, buckets, and jerrycans (5–25L), and 3) drums (>25L). "Other" purpose included bating, cooking, drinking, and multiple functions. The total number of immature mosquitoes and habitats varied by arm and time.

In the intervention group, there was a significant reduction in the percentage of immature mosquitoes in animal water containers (11.58%). However, there was an increase in mosquito breeding in jerrycans (4-.24%) and drums (32.04%). Containers used for other purposes (e.g., bating, cooking, drinking) remained significant breeding sites, accounting for 71.69% of immature mosquitoes (Fig 1).

There was significant difference in the distribution of immature mosquitoes among different container types ($\chi2 = 12.45$, $p = 0.02$) and container purposes ($\chi2 = 10.78$, $p = 0.03$) between the control and intervention groups at baseline. There was also a strong association between the presence of jerrycans and the number of immature mosquitoes ($r = 0.62$, $p < 0.01$) in the intervention group at 12 months (Fig 1).

## Discussion

This study aimed to evaluate the effectiveness of a mixed school- and community-based educational intervention on *Aedes aegypti* control behaviors, focusing on arboviral knowledge, attitudes, and behaviors related to larval source reduction. The findings provide important insights into the potential of educational intervention to influence mosquito control practices at the household level.

Our study revealed significant improvements in knowledge, attitudes, and practices among children in the intervention group compared to the control group. At the 12-month follow-up, children in the intervention group demonstrated increased awareness of source reduction techniques and how to prevent the transmission of mosquito-borne diseases. These results align with previous studies indicating that educational interventions can effectively increase mosquito control behaviors [32–34]. The improvement in self-reported practices such as turning over unused water containers, highlights the role of children as agents of change within their households, and understanding the relationship between household perceptions and children's ability to make change is crucial for designing effective, targeted interventions.

The house index, which measured the number of houses with at least one productive container, increased from baseline to the 12-month follow-up in both the intervention and control groups (Table 1). The increase in house index in both groups indicates that while the intervention improved knowledge and some behaviors, additional challenges remain in eliminating productive breeding sites. This rise could be due to several factors, including seasonal rainfall variation,

difference in adherence to intervention recommendations, or the natural resiliency of mosquito breeding habitats [43]. Nonetheless, this finding highlights the dynamic nature of mosquito breeding sites and the importance of continuous monitoring and targeted interventions to address evolving challenges in vector control.

Our vector data identified an increase in the percentage of immature mosquitoes in jerrycans and drums at the 12-month follow-up in the intervention group, an unexpected finding that was not observed in the control group. This could be attributed to the fact that jerrycans and drums are often used for water storage and are frequently left uncovered due to their large capacity, so this finding clearly underscores the need for continued community education on the importance of covering water storage containers. Additionally, seasonality may play a role, as the surveys were solely conducted during the rainy season, potentially leading to higher water accumulation and mosquito breeding in these containers.

The chi-square tests revealed significant differences in the distribution of immature mosquitoes among different container types and purposes between the control and intervention groups at 12 months. Specifically, the intervention group showed a significant reduction in the percentage of immature mosquitoes in animal water containers, but there was also an increase in mosquito breeding in jerrycans and drums. Correlation analysis further supported these findings by demonstrating a strong association between the presence of jerrycans and the number of immature mosquitoes in intervention households at 12 months. This highlights the importance of targeting specific container types in future intervention to achieve better vector control.

Overall, the intervention revealed that while some source reduction behaviors, such as covering water containers, were widely adopted, others such as removing non-purpose containers were not as effectively implemented, as seen by the increase in the number of immature mosquitoes within no-purpose containers (Fig 1). This indicates that certain behavioral recommendations align better with existing hygiene norms and practices, and in order to best enhance the effectiveness of source reduction interventions, it is important to address the community's perceptions and practices regarding container management [39]. For example, the alignment of source reduction behaviors with existing hygiene practices, such as keeping drinking water clean, was a key factor in the successful adoption of certain behaviors. However, certain practices such as managing no-purpose containers faces resistance, suggesting that interventions should be tailored to fit within the community's existing practices to achiever broader acceptance and implementation [39].

Conducting the intervention in an urban setting posed unique challenges compared to similar efforts in rural areas [29]. Urban environments often have higher population densities, increased movement of people, and greater variability in container usage and waste management practices [19]. These factors complicate efforts to implement and sustain source reduction techniques, but at the same time, urban settings also offer opportunities for broader reach and the potential to integrate interventions with existing public health infrastructure. Addressing the specific challenges of urban areas such as informal settlements and inadequate waste management systems requires careful consideration and involvement of local authorities and community leaders to ensure that the intervention is properly implemented and adopted by the people.

## Limitations

One limitation of this study is that the entomological surveys were conducted solely in the rainy season and may not properly reflect the breeding patters throughout the year. Also, there was considerably more rainfall from October to December 2019, which may have influenced the entomological indices at the 12-month follow-up compared to baseline [44]. Future studies should incorporate longitudinal data collection to account for seasonal variation.

Another limitation of this study was the inability to measure the direct impact of the intervention on reducing mosquito abundance and arboviral disease incidence. Future research and work should explore these outcomes in more detail, in order to see how the educational intervention can produce a change at a community health level.

## Conclusion and future research

The findings of this study contribute to the existing literature by demonstrating the long-term effectiveness of a combined school and household-based educational intervention in improving knowledge, attitudes, and practices related to *Aedes*

*aegypti* control. This study highlights the importance of integrating educational programs into public health strategies to combat mosquito-borne diseases. Targeting both children at school and caregivers in their house led to a more holistic approach by addressing multiple layers of influence including socioeconomic status and access to resources.

Future research should focus on the long-term sustainability of these behaviors and explore additional strategies to mitigate the limitations identified. For example, coordinated community clean-up events and engagement with local leaders could enhance the effect of source reduction efforts [28].

Ultimately, this study highlights the potential of educational interventions to bring about meaningful changes in public health, particularly in underserved and at-risk communities. By empowering individuals with knowledge and practical skills, such interventions can contribute to the prevention of mosquito-borne illnesses, improving the health outcomes and quality of life in affected regions,

## Supporting information

**S1 Fig. Infographic with the different components of the educational curriculum.** This infographic summarizes the key educational messages related to vector-borne disease prevention, environmental sanitation, personal hygiene, and community engagement.
(TIFF)

**S2 Fig. Consort chart for village selection and collected data from caregivers.** Flowchart describing the number of caregivers included at each stage of the analysis. Exclusion and attrition numbers at each stage are detailed.
(TIFF)

**S3 Fig. Consort chart for village selection and collected data from children.** Flowchart describing the number of children included at each stage of the analysis. Exclusion and attrition numbers at each stage are detailed.
(TIFF)

**S1 Table. Baseline KAP for caregivers.**
(XLSX)

**S1 Text. Inclusivity in global research.**
(DOCX)

## Acknowledgments

We would like to express our gratitude to the students and their families for the participation in the study. We also would like to thank the teachers for allowing us to use their class time and helping with implementing the educational curriculum. We also want to express our gratitude to the field team for their hard work and dedication while conducting the house visits, monitoring mosquito populations, and engaging with the community. Finally, we want to thank the Likoni Ministry of Health for allowing us to conduct this study and their contributions to the development of the intervention curriculum.

## Author contributions

**Conceptualization:** Jenna Forsyth, Mwashee Lutt, Angelle Desiree LaBeaud.

**Data curation:** Prathik Kalva, Francis Mutuku, Gladys Agola, Mwashee Lutt, Angelle Desiree LaBeaud.

**Formal analysis:** Prathik Kalva.

**Investigation:** Jenna Forsyth, Francis Mutuku, Gladys Agola, Mwashee Lutt, Angelle Desiree LaBeaud.

**Methodology:** Jenna Forsyth, Angelle Desiree LaBeaud.

**Project administration:** Francis Mutuku, Gladys Agola, Mwashee Lutt, Angelle Desiree LaBeaud.

**Resources:** Jenna Forsyth.

**Supervision:** Angelle Desiree LaBeaud.

**Visualization:** Prathik Kalva.

**Writing – original draft:** Prathik Kalva.

**Writing – review & editing:** Prathik Kalva, Jenna Forsyth.

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
