## [Decision Letter · Decision Letter 0]

4 Nov 2024

PGPH-D-24-01687

School and Home-Based Educational Intervention in Urban Kenya: Sustained Improvements in Knowledge, Attitudes, and Behaviors for Aedes aegypti Control

Dear Dr. Kalva,

Thank you for submitting your manuscript to PLOS Global Public Health. After careful consideration, we feel that it has merit but does not fully meet PLOS Global Public Health’s publication criteria as it currently stands. Therefore, we invite you to submit a revised version of the manuscript that addresses the points raised during the review process.

EDITOR: Many thanks for your submission on the pertinent topic of dengue in Africa. Please could you address the comments from the reviewers' in order for us to consider acceptance.

We look forward to receiving your revised manuscript.

Kind regards,

Xin Hui Chan

Academic Editor

Journal Requirements:

Additional Editor Comments (if provided):

Reviewers' comments:

Reviewer's Responses to Questions

**Comments to the Author**

1. Does this manuscript meet PLOS Global Public Health’s publication criteria ? Is the manuscript technically sound, and do the data support the conclusions? The manuscript must describe methodologically and ethically rigorous research with conclusions that are appropriately drawn based on the data presented.

Reviewer #1: Yes

Reviewer #2: Partly

2. Has the statistical analysis been performed appropriately and rigorously?

Reviewer #1: No

Reviewer #2: No

3. Have the authors made all data underlying the findings in their manuscript fully available (please refer to the Data Availability Statement at the start of the manuscript PDF file)?

Reviewer #1: No

Reviewer #2: No

4. Is the manuscript presented in an intelligible fashion and written in standard English?

Reviewer #1: Yes

Reviewer #2: No

5. Review Comments to the Author

Reviewer #1: General Comment:

- The authors reported an important topic of Aedes aegypti control. It’s a caregiver-targeted household-based intervention combined with a school-based educational intervention to build knowledge, skills, and create a sense of self efficacy in a dengue-endemic area of urban Kenya. The study is well designed, but the review has some concerns on the data reported.

Major Comment:

- The authors reported the detail KAB of children as “agents of change” and caregivers’ responses to the questionnaires were not analysed due to differences at baseline. The reviewer is not clear what is meant by the “differences at baseline” and how was that assessed. I propose the authors consult a statistician how to analyse KAB of children as well as caregivers for the entomological indices as outcome.

Detail comments:

Introduction:

- Reference 25 and 26 are repeated. Please cite the correct paper for Ref 26.

Methods

- Please briefly mention why Likoni was selected and if available, provide dengue prevalence data.

- Expand the description on pilot, rather than simply citing reference 24.

- Villages were selected. Please add a statement, why they are considered as Urban communities?

- Add about ‘Fourth grade student’ this section. Currently, Fourth grade was mentioned in abstract and results only.

- Are there any implications for recruiting less/ analysing less than the intended numbers?

- Were there any differences in the characteristics of the children who completed the survey and who did not?

- If possible, add Consort flow chart for clustered RCTs

- How many trainers were there in home training sessions? Do they speak the same language as the caregivers?

- Describe how the scoring of the KAB were done?

Results

- Please confirm the numbers of caregivers were involved in data collection and the number used data analysis are the same (i.e., 492). If any caregivers were excluded because of missing data, please edit the paragraph accordingly.

- Similarly, please describe how many children were recruited and re-confirm how many children were analysed (i.e., 445)

Table 1

- Table 1 shown for the numbers of caregivers (i.e., 243 vs 249). If the main intention was to report KAB of children, will authors consider to report the Table 1 by the numbers of children instead (i.e., 217 vs 228)?

- Were there any statistical tests performed to assess any differences in the two groups?

- Numbers were missing for female and marital statuses.

Table 2

- Table 2, add p-values comparing baseline data.

- Edit the table title to reflect the reporting of Entomological indices in Table 2

Table 3

- Table 3 is missing in the submission/ review.

Discussion

- Add reference for third and fifth paragraphs (i.e. The house index… Overall…).

- Entomological indices supposed to validate the self-reported measure. Therefore, self-reported measure should not be listed as a limitation.

Conclusion

- The second sentence was not supported by the findings from the study (i.e., The findings underscore ….). Please revise.

- Conclusions should not introduce new agenda. Please clarify why the need for solid waste management was mentioned in conclusion?

Reviewer #2: General: Lastly, and most importantly, the authors should clearly demonstrate the novel findings of their study in the body of literature. This could be achieved by highlighting how their study adds to existing literature and identifying any new or significant findings.

Introduction: When it comes to a review of the literature, the introduction is quite brief. Information about dengue and other arboviruses' morbidity or morality, their prevalence globally or in Sub-Saharan Africa, who is more impacted, and why should be included. The authors may take advantage of the extensive body of research on arboviruses and the role children tackle them. The authors might offer a list of additional potential interventions in addition to school-based. Lastly, other than examining knowledge, attitudes, or behavioral changes, what are the specific guiding research questions/hypotheses for this study?

Methods section: I am unclear why the authors included children in the KAP assessment. Also, not clear how the questions administered were standardized and validated? Authors need to define what increases or decrease in knowledge, attitude, or behavioral change. How were these variables measured? Analysis should consider additional statistical tests beyond descriptive statistics.

6. PLOS authors have the option to publish the peer review history of their article (what does this mean? ). If published, this will include your full peer review and any attached files.

**Do you want your identity to be public for this peer review?** For information about this choice, including consent withdrawal, please see our Privacy Policy .

Reviewer #1: No

Reviewer #2: **Yes: ** Tewelde Tesfaye Gebremariam

---

## [Editor Report · Decision Letter 1]

1 Apr 2025

PGPH-D-24-01687R1

School and Home-Based Educational Intervention in Urban Kenya: Sustained Improvements in Knowledge, Attitudes, and Behaviors for Aedes aegypti Control

Dear Dr. Kalva,

Thank you for submitting your manuscript to PLOS Global Public Health. After careful consideration, we feel that it has merit but does not fully meet PLOS Global Public Health’s publication criteria as it currently stands. Therefore, we invite you to submit a revised version of the manuscript that addresses the points raised during the review process.

We look forward to receiving your revised manuscript.

Kind regards,

Xin Hui Chan

Academic Editor

Journal Requirements:

Additional Editor Comments (if provided):

Many thanks for your revised submission with detailed updates and responses.

Grateful if you could attend to some very minor typographical errors for acceptance.

"Line 40: missing p value in parentheses

Line 569: 'at 12 months' not 'as 12 months'"
---

## [Editor Report · Decision Letter 2]

8 Apr 2025

School and Home-Based Educational Intervention in Urban Kenya: Sustained Improvements in Knowledge, Attitudes, and Behaviors for Aedes aegypti Control

PGPH-D-24-01687R2

Dear Mr Kalva,

We are pleased to inform you that your manuscript 'School and Home-Based Educational Intervention in Urban Kenya: Sustained Improvements in Knowledge, Attitudes, and Behaviors for Aedes aegypti Control' has been provisionally accepted for publication in PLOS Global Public Health.

Best regards,

Xin Hui Chan

Academic Editor